# Metformin Treatment Regulates the Expression of Molecules Involved in Adiponectin and Insulin Signaling Pathways in Endometria from Women with Obesity-Associated Insulin Resistance and PCOS

**DOI:** 10.3390/ijms23073922

**Published:** 2022-04-01

**Authors:** Maria Lorena Oróstica, Isis Astorga, Francisca Plaza-Parrochia, Cristian Poblete, Rodrigo Carvajal, Víctor García, Carmen Romero, Margarita Vega

**Affiliations:** 1Laboratorio de Fisiopatología Celular y Cáncer (FICEC), Centro de Investigación Biomédica (CIB), Facultad de Medicina, Universidad Diego Portales, Santiago 8370007, Chile; 2Laboratorio de Comunicaciones Celulares, Programa de Biología Celular y Molecular, Centro de Estudios en Ejercicio, Metabolismo y Cáncer (CEMC), Facultad de Medicina, Universidad de Chile, Santiago 8380453, Chile; 3Laboratory of Endocrinology and Reproductive Biology, Clinical Hospital, University of Chile, Santos Dumont #999, Santiago 8380456, Chile; isispaulina@gmail.com (I.A.); franciscaplazaparrochia@gmail.com (F.P.-P.); cr.pobleterivas@gmail.com (C.P.); dr.rodrigocarvajal@gmail.com (R.C.); vicrodrigo73@gmail.com (V.G.); cromero@hcuch.cl (C.R.); 4Department of Obstetrics and Gynecology, School of Medicine, Clinical Hospital, University of Chile, Santiago 8380456, Chile

**Keywords:** metformin, adiponectin, insulin resistance, obesity, PCOS, endometrium

## Abstract

Polycystic ovary syndrome (PCOS) is an endocrine/metabolic disorder associated with insulin resistance (IR) and obesity. Endometria from women with PCOS present failures in insulin action, glucose uptake and signaling of insulin-sensitizing molecules, such as adiponectin, with consequences for reproduction. Metformin (MTF) treatment improves insulin signaling in endometrial tissues, but its mechanism is not fully understood. This study addresses the MTF effect, as well as adiponectin agonist action, on levels of molecules associated with insulin and adiponectin signaling pathways in endometrial tissue and cells, as assessed by immunohistochemistry and immunocytochemistry, respectively. Endometrial tissues were obtained from women and divided into five groups: Normal Weight (control); Obesity + IR; Obesity + IR + PCOS; Obesity + IR + MTF; Obesity + IR + PCOS + MTF. Endometrial cells stimulated with TNFα (as obesity-marker) were also used to partially emulate an obesity environment. The results showed low levels of insulin/adiponectin signaling in the endometria from women with obesity, IR and PCOS compared with the control group. MTF re-established these levels, independently of PCOS. TNFα-associated molecules were elevated in pathologic endometria, whereas MTF diminished these levels. The low levels of insulin/adiponectin molecules in endometrial cells treated with TNFα were reverted by MTF, similar to what was observed in the case of the adiponectin agonist. Therefore, independently of PCOS, MTF can re-establish levels of molecules involved in insulin/adiponectin signaling in endometrial cells, suggesting an improvement in insulin action and reproductive failures observed in endometria from women with obesity/PCOS.

## 1. Introduction

Polycystic ovary syndrome is an endocrine and metabolic disorder that affects between 5 and 18% of women of childbearing age [1,2]. This syndrome is characterized by hyperandrogenism and about 80% of cases also present metabolic alterations, such as obesity and hyperinsulinemia [1].

These women present reproductive alterations, such as infertility, abortions and recurrent abortions [3]. In this regard, the endometrial tissues of these women present metabolic alterations that may explain, in part, the reproductive problems observed in these patients [4,5]. Likewise, in endometria from women with a high body mass index (BMI) and PCOS, insulin-mediated glucose uptake machinery is altered, characterized by a lower expression of GLUT4 insulin-dependent glucose transporters, phosphorylation of IRS1 and AS160 and protein levels of the WAVE family, which leads to a lower translocation of GLUT4 vesicles from the intracellular compartment to the cellular plasma membrane [4,6]. Additionally, it has been shown that, as a consequence of these alterations, the endometrium has a lower ability to capture glucose, as observed in endometrial cells exposed to hyperinsulinemic and hyperandrogenic conditions that resemble PCOS and obesity environments [7]. Therefore, PCOS and obesity conditions favor metabolic alterations in tissues, which may explain the failures in certain processes, e.g., implantation, observed in these women. These effects could be the result of an imbalance in obesity markers that regulate the action of insulin in target tissues, such as the endometrium [8]. In fact, in endometria from women with obesity and PCOS, increased levels of molecules related to the tumor necrosis factor-alpha (TNFα) signaling pathway, which is considered to be an obesity marker and negative regulator of insulin action, can be observed [7,8]. Additionally, it has been shown that TNFα not only negatively affects the action of insulin but also can affect the signaling of insulin-sensitizing molecules, such as adiponectin in human endometrial cells [9,10].

Adiponectin is a protein hormone mainly secreted by adipose tissue [11]. It exhibits insulin-sensitizing activity, increases skeletal muscle glucose uptake and inhibits the action of TNFα [11,12,13,14]. On the other hand, in conditions of insulin resistance (IR) and inflammation, both conditions typical of obesity, the protein expression of adiponectin decreases [9,10,15,16,17].

The insulin and adiponectin signaling pathways converge on the APPL1 adapter protein [14]. APPL1 is an adaptor protein containing a pleckstrin homology (PH) domain, a phosphotyrosine-binding (PTB) domain and leucine zipper motif 1. The binding of adiponectin and insulin to their receptors, triggers the coupling of the receptors with the APPL1–IRS1/2 complex. This binding favors the activation of the insulin pathway. In obesity conditions, APPL1 interaction with insulin receptors is diminished [18], whereas the interaction between APPL1 with adiponectin receptors promotes activation of AMP-activated kinase (AMPK), favoring insulin sensitivity [17,18]. In addition, it has been determined that activation pathways through APPL1 increase the translocation of GLUT4, favoring glucose uptake [18,19]. Likewise, APPL1 is able to activate the phosphorylation of Akt, which also promotes the interaction between insulin receptors and IRS1/2 [18]. Meanwhile, APPL2 has opposite effects to those of APPL1, decreasing glucose uptake stimulated by insulin [20]. The mechanisms by which APPL2 exert its action are not completely clear; some antecedents indicate that APPL2 binds to APPL1, blocking adiponectin signaling. Similarly, APPL2 could negatively affect Akt action [20]. It has been reported that APPL2 decreases insulin-mediated phosphorylation of AS160, preventing GLUT4 from being translocated to the cell membrane [18,21,22,23]. Additionally, it has been determined that serum levels of adiponectin and endometrial levels of APPL1 in patients with PCOS with obesity, hyperandrogenism and hyperinsulinism are diminished [9]. In vitro studies with endometrial cells under conditions of high levels of testosterone and insulin confirm the failures in the signaling pathway of adiponectin, characterized by low levels of adiponectin, its receptors and APPL1 [9].

As is known, metformin (MTF) is a drug widely used to reduce hyperglycemia in type 2 diabetic patients [17,24]. This biguanide can increase insulin sensitivity in skeletal muscle and the liver [24]. One of its mechanisms of action is to increase the translocation of the glucose transporters GLUT1 and GLUT4 to the cell membrane [17]. GLUT4, in particular, is expressed more abundantly in the cell membrane with an increase of phosphorylated AS160 levels and the translocation of vesicles with GLUT4 to the plasma membrane [24,25]. It has also been reported that MTF may promote APPL1 separation from APPL2, and thus prevent the repressive effect of APPL2 on APPL1 [26]. Studies in patients with PCOS and obesity treated with MTF have shown that, at the endometrial level, the use of MTF increases the phosphorylation of AMPK and, consequently, the transcription factor MEF2A, thus regulating the expression of GLUT4 by increasing the transcription and translation of this transporter [17]. However, it is unknown whether MTF can re-establish the levels of adiponectin-associated molecules, such as APPL1 or APPL2, among others, and consequently enhance insulin action in pathological endometrial tissue. In this work, we evaluated the effect of MTF and the effect of an adiponectin analog as another insulin-sensitizing agent on the expression of molecules associated with adiponectin and insulin signaling pathways in endometrial tissue and endometrial cells under obesity conditions.

## 2. Materials and Methods

### 2.1. Subjects and Design

This is a case–control study with five experimental groups: endometria obtained from women with Normal Weight (control group) (*n = 7*), Obese + IR (*n = 7*), Obese + IR + PCOS (*n = 7*), Obese + IR + MTF (*n = 7*) and Obese + IR + PCOS + MTF (*n = 7*). Clinical and endocrine parameters of the patients are shown in Table 1. The tissue samples were obtained with a Pipelle suction curette from the corpus of the uteri from all participants. All endometrial samples were obtained at the time of hysterectomy due to benign uterine pathology. The diagnosis of PCOS was made according to the Androgen Excess and PCOS Society [2]. Hyperprolactinemia, androgen-secreting tumors, Cushing’s syndrome, congenital adrenal hyperplasia, attenuated 21-hydroxylase deficiency, as well as thyroid disease, were excluded by appropriate tests. Patients with obesity (with or without PCOS) previously diagnosed as hyperinsulinemic and IR by HOMA-IR [27] and ISI Composite indexes [28] were treated with MTF (850 mg twice a day for at least 12 weeks) at the time of the recruitment (Obesity + IR + MTF and Obesity + IR + PCOS + MTF groups). Endometrial tissues were obtained from all women in the study groups during the proliferative phase of the menstrual cycle. This investigation was approved by the School of Medicine and Clinical Hospital Ethical Committees of the University of Chile, and informed written consent was obtained from all subjects. None of these women had taken oral contraceptives or other medications (except for MTF in the IR groups) for at least 3 months before starting the study. Based on histological dating and classification according to Noyes criteria [29] as carried out by an experienced pathologist, all the endometrial samples were selected in the proliferation phase.

### 2.2. Materials and Reagents

The utilized antibodies for immunohistochemistry and immunocytochemistry were APPL1 (#3858 Cell Signaling, Danvers, MA, USA; antibody dilution: 1:100), APPL2 (#14294-1-AP Thermo Fisher Scientific, Waltham, MA, USA; antibody dilution: 1:200), α-p-p38-MAPK (T180/Y182) (#4511 Cell Signaling, Danvers, MA, USA; antibody dilution: 1:100), NF-κB (p65) (#8242 Cell Signaling, Danvers, MA, USA; antibody dilution: 1:500), GLUT4 (#sc-53566 Santa Cruz Biotechnology, Dallas, TX, USA; antibody dilution: 1:200), p-AS160 (T642) (#MBS002191 MyBiosource, San Diego, CA, USA; antibody dilution: 1:100), p-MEK (S189) (#ab194809 Abcam, Cambridge, UK; antibody dilution: 1:200), anti-IgG Rabbit secondary antibody (#31460 Cell Signaling, Danvers, MA, USA) and anti-IgG Mouse secondary antibody (#31430 Cell Signaling, Danvers, MA, USA). In this study, other reagents for stimulated cells in in vitro assays were used: an adiponectin agonist called AdipoRon (AdipoR agonist, 2-(4-Benzoylphenoxy)-*N*-[1-(phenylmethyl)-4-piperidinyl]-acetamide, #SML0998, Sigma-Aldrich (Merck), Darmstadt, Germany); a human recombinant protein called TNFα (tumor necrosis factor alpha, #H8916, Sigma-Aldrich, MO, USA) and metformin (1,1-Dimethylbiguanide hydrochloride, #1115-70.4, Sigma-Aldrich (Merck), Darmstadt, Germany).

### 2.3. Immunohistochemistry

This technique was used to assess the localization and semi-quantitation of total GLUT4, APPL1, APPL2 and NF-κB, and phosphorylated AS160 in T642 and phosphorylated p38-MAPK in T180/Y182. Immunostaining was performed on 5 μm sections of formalin-fixed paraffin-embedded ovarian biopsies. Briefly, tissue sections were deparaffinized in xylene and hydrated in a series of graded alcohols. The sections were incubated in an antigen retrieval solution (10 mM sodium citrate buffer, pH 6) at 95 °C for 20 min. Endogenous peroxidase activity was prevented by incubating the samples in 3% hydrogen peroxide for 5 min. Nonspecific antibody binding was blocked with BSA/PBS at 2%. The endometria sections were incubated for 18 h at 4 °C with specific primary antibodies. Negative controls were analyzed on adjacent sections incubated without the primary antibody. The slides were incubated for 20 min with the biotinylated anti-rabbit secondary antibody (1:300). The reaction was developed by the streptavidin–peroxidase system, and DAB (3-3′ diaminobenzidine) was used as the chromogen; counterstaining was carried out with hematoxylin. The slides were evaluated in a Nikon optical microscope (Nikon Inc., Melville, NY, USA). Six images within the field of vision per each tissue sample were evaluated. Each sample was evaluated by the Imagen Pro Plus 6.1 computer program, measuring integrated optical density (IOD) expressed as arbitrary units (AU). The results were expressed as the mean (standard error of the mean, SEM) per study group of women.

### 2.4. Cell Culture and Treatments

The cell line St-T1b was utilized in the present investigation. This cell line was obtained from stromal cells of endometrial biopsies in the proliferative phase of the menstrual cycle of normal premenopausal women [30]. The cells were cultured in DMEM/Ham F12 medium (Sigma Aldrich Co., St. Louis, MO, USA) with 10% fetal bovine serum treated with dextran carbon (Hyclone; Thermoscientific, New York, NY, USA) and 1× antimycotic/antibiotic at 37 °C in a 5% CO_2_ atmosphere until 80% confluence. Then, cells were further cultured directly on 4-well slides (Thermoscientific Nunc Lab-Tek II Chamber Slide System; Thermo Fisher Scientific, New York, NY, USA; 60,000 cells/well) in growth media for 24 h at 37 °C in 5% CO_2_/atmosphere for immunocytochemistry (ICC). Cultures were then washed twice with sterile Dulbecco’s Phosphate Buffered Saline (PBS; GIBCO; Invitrogen Corporation, Camarillo, CA, USA) and further treated with TNFα (100 ng/mL/well) for 24 h in serum-free medium. Other cell cultures were co-stimulated with TNFα (100 ng/mL) plus MTF (20 nM) for 24 h in serum-free medium to determine TNFα and MTF effects on protein expression levels by immunocytochemistry (ICC). In addition, other cells were stimulated with TNFα plus AdipoRon (30 µM) for 24 h in serum-free medium to determine the protein levels by ICC. AdipoRon is an agonist of adiponectin, which activates the adiponectin pathway by increasing phosphorylation that activates AMPK, increases insulin sensitivity and decreases pro-inflammatory cytokines [31]. The reagent concentrations used to stimulate the cells were obtained from dose–response curves for each reagent. The basal condition corresponded to cultures with no stimulation (serum-free medium only). Negative controls were analyzed on cells incubated without the primary antibody. Three independent experiments were performed for each treatment (*n* = 3; in duplicate). The TNFα experimental condition was previously used in Oróstica et al. 2018 [10].

### 2.5. Immunocytochemistry

Briefly, St-T1b cells were fixed with a solution containing 4% paraformaldehyde in PBS for 15 min at room temperature. The endogenous peroxidase activity was inhibited by incubation in 3% H_2_O_2_ for 15 min and nonspecific binding was blocked with PBS–bovine serum albumin (BSA) 2% for 10 min. Cells were incubated with the primary antibody overnight at 4 °C: APPL1, APPL2, p-AS160 (T642), p-p38-MAPK (T180/Y182), IRS1, p-IRS1(Y612) and p-IRS1(S270) antibodies. Then, cells were washed with PBS 1× solution 3 times and incubated with a horseradish peroxidase (HRP)-labeled secondary antibody to rabbit or mouse immunoglobulin G (IgG; KPL, Gaithersburg, MD, USA; dilution 1:300) for 2 h at 37 °C. Chromogenic marks were revealed using 3,30-diaminobenzidine as substrate, and counterstaining was performed with hematoxylin for 5 s. Six images within the field of vision per each condition were evaluated. The analysis was performed by the measurement of positive pixel intensity with the use of the semiquantitative analysis tool-integrated optical density (IOD) of the Image-Pro Plus 6.2 software. The data are presented as IOD arbitrary units (AU). The results were expressed as the mean (standard error of the mean, SEM) per culture condition.

### 2.6. Statistical Evaluation

The number of subjects per group was calculated assuming α = 0.05 and β = 0.2, a difference between means of 0.25 and standard deviation according to our previous studies [4,6,7,9,17,32]. Comparisons between groups were performed using the Kruskal–Wallis test and Dunn’s post hoc test. Statistical tests were performed using Stata 9 and GraphPad Prism 6.0. In the in vitro assays, the experiments were performed three times in duplicate for each experimental condition. *p*-values < 0.05 were considered statistically significant.

## 3. Results

### 3.1. Effect of MTF on Molecules Associated with Insulin Action in Endometria from Women with IR, Obesity and PCOS

As has already been mentioned, our previous reports showed diminished levels of molecules associated with insulin signaling in endometria from women with obesity (plus IR) and PCOS [4,6,7,17,32,33]. In addition, GLUT4 protein levels were partially reestablished in this tissue after MTF treatment [17]. In the present investigation, it was evaluated whether the effect of MTF in the endometrium is different when women present obesity-associated IR in the absence of PCOS.

The results show that AS160 phosphorylated in threonine 642 residue decreased in endometria from women with obesity and women with obesity plus PCOS (both with IR) compared with the Normal Weight group (control group) (Figure 1). Interestingly, MTF treatment induced an increase of phospho-AS160 endometrial levels compared with the group of women without treatment (*p* < 0.05) (Figure 1A). Regarding GLUT4, the results indicate that MTF had the strongest effect on levels of this protein. Figure 1B shows that the obesity significantly decreased GLUT4 levels in the endometrium under the IR condition; this finding was observed in women with or without PCOS, whereas MTF oral treatment was able to elevate GLUT4 levels in the endometrium under the IR state (*p* < 0.05). Even more, MTF treatment induced levels of GLUT4 higher than the levels observed in the control group (Figure 1B). It is most likely that the effect of MTF at the endometrial level is independent of the PCOS condition.

### 3.2. Effect of MTF on Molecules Associated with Adiponectin Signaling in Endometria from Women with IR, Obesity and PCOS

Previously, we have shown that levels of molecules associated with adiponectin signaling are decreased in endometria from women with obesity-associated IR and PCOS [9] compared with the group of women with obesity only, suggesting a potential lower insulin-sensitizing action of adiponectin in this tissue. Therefore, we evaluated whether MTF could restore the levels of adiponectin signaling molecule APPL1 and/or lower APPL2 levels.

The data of the present study indicate that endometrial APPL1 levels were lower in women with obesity and obesity plus PCOS (both with IR condition, *p* < 0.05) (Figure 2A). On the other hand, the endometrial APPL2 levels were higher in these same groups (Figure 2B) compared to the control group (*p* < 0.05). Interestingly, MTF oral treatment in both groups of women (Obese + IR and Obese + IR + PCOS groups) induces an increase of APPL1 and a decrease of APPL2 levels in the endometrium (*p* < 0.05 compared with the control group), independently of the PCOS condition.

### 3.3. Effect of MTF on the Active Form of p38-MAPK and NF-κB Levels in Endometria from Women with Obesity-Associated IR and PCOS

Previously, our group has reported that a pro-inflammatory environment is present in endometria from women with obesity and with obesity plus PCOS [7]. This inflammatory state is characterized by macrophages, high levels of TNF-α and nuclear NF-κB and lower levels of adiponectin protein [7], this being one of the probable causes of failure in endometrial function in these women. Therefore, in this study the effect of MTF treatment on the activation of MAPK related to TNFα action was assessed by evaluating the active form of p38-MAPK (phospho-p38-MAPK (T180/Y182) levels and nuclear NF-κB levels [34]. Figure 3 shows that p-p38-MAPK levels were lower in endometria from women with obesity + IR and obesity + IR + PCOS after MTF treatment compared to all other groups (*p* < 0.05) (Figure 3A). Regarding NF-κB, the data show that MTF can lower the nuclear presence of NF-κB in endometrial tissue from women with obesity + IR and obesity + IR + PCOS compared with the control group or women without treatment (*p* < 0.05) (Figure 3B).

### 3.4. Metformin Re-Established Levels of Insulin and Adiponectin Signaling Molecules in Cultured Human Endometrial Cells

Until now, the results of the present work have shown that molecules associated with the insulin and adiponectin signaling pathways are decreased in women with obesity independently of the PCOS condition; moreover, these levels could be re-established with MTF treatment. In addition, in previous studies, it has been observed that TNFα, an obesity marker related to the obesity-associated inflammatory state, reduces protein levels of GLUT-4, adiponectin and phospho-AMPK, besides reducing glucose uptake by stromal endometrial human cells [7,10]. In the present investigation, it was evaluated whether MTF is able to re-establish levels of phospho-AS160, APPL1, APPL2 and p-MEK (S189) in endometrial cells treated with TNFα. The results show that TNFα reduces p-AS160 levels compared with the basal condition (*p* < 0.05), although the addition of MTF to cell cultures was able to re-establish these levels in an in vitro cell model (*p* < 0.05 vs. TNFα condition) (Figure 4). In addition, MTF increased the levels of molecules associated with the adiponectin signaling pathway (APPL1, APPL2 and p-MEK-S189) that were reduced by the effect of TNFα (*p* < 0.05 compared to the TNFα condition) (Figure 5).

### 3.5. Effect of Adiponectin Agonist on Molecules Associated with Insulin Signaling in Human Endometrial Cells

Reports from our group showed that adiponectin signaling molecules are decreased in endometrial tissues from women with obesity and/or PCOS [9], which is also observed in cultured human endometrial cells treated with TNFα or with insulin plus testosterone, characteristic of obesity and PCOS, respectively [10]. In this work, it was evaluated whether adiponectin has a similar effect as MTF on insulin signaling molecules in human endometrial cells by using a synthetic small-molecule agonist of the adiponectin receptors called AdipoRon. First, the effects of the obesity-related molecule TNFα and the insulin-sensitizer AdipoRon on molecules associated with the adiponectin signaling pathway in the in vitro model were evaluated. The results showed that TNFα decreases APPL1 levels, whereas an increase in APPL2 levels was observed (*p* < 0.05 vs. basal). Interestingly, AdipoRon treatment was capable of re-establishing these levels to basal conditions (Figure 6).

Furthermore, in this in vitro cell model, certain molecules associated with the insulin signaling pathway were also examined. Figure 7A shows that AdipoRon returned to basal conditions the TNFα-reduced p-AS160 and IRS1 levels in human endometrial cells (*p* < 0.05 vs. TNFα condition). In addition, we evaluated the effect of AdipoRon on levels of active and inactive forms of IRS1 (p-IRS1-Y612 and p-IRS1-S270, respectively) in human endometrial cells treated with TNFα. The results show that AdipoRon treatment induced an increase in the active IRS1 (p-IRS1-Y612) levels compared to cells treated with TNFα (*p* < 0.05) (Figure 7B). Nevertheless, the levels of the inactive form of IRS1 were unchanged despite the treatment used (*p* > 0.05) (Figure 7B). These results suggest that adiponectin could act as an insulin-sensitizer, improving insulin action in endometrial cells in an obesity environment.

## 4. Discussion

Previous studies carried out by our group have shown that protein levels of molecules involved in the adiponectin signaling pathway are decreased in endometrial tissue from women with obesity and PCOS. This is a relevant aspect considering that adiponectin has an important insulin-sensitizing action in several tissues. In addition, these alterations in adiponectin signaling are also observed in an in vitro cell system where human endometrial cells are cultured under hyperandrogenic and hyperinsulinemic conditions and treated with TNFα. As previously reported, these experimental conditions resemble the PCOS condition, IR and obesity [9,10].

In the present study, it was observed that in the endometria from women with obesity and IR (with or without PCOS), lower levels of p-AS160 and GLUT4 were detected and that these levels increased after MTF treatment, which agrees with the study of Carvajal et al., 2013 [17]. Moreover, it was observed that key downstream regulatory molecules associated with adiponectin signaling, such as APPL1 and APPL2, are altered in endometria from women with obesity and IR, with or without PCOS. Importantly, MTF oral treatment was capable of restoring the levels of these molecules to basal conditions, suggesting a positive effect of MTF in insulin and adiponectin signaling pathways in human endometrial cells. Probably, the effect of MTF in women with PCOS is mainly related to the improvement of inflammatory parameters associated with obesity. In this regard, one study showed that MTF increases circulating adiponectin and decreases serum TNFα and C-reactive protein levels in women with PCOS with an elevated BMI [35]. Furthermore, other studies indicate that MTF changes the expression and translocation of molecules associated with the adiponectin signaling pathway, increasing not only the availability of the ligand but also the activation of this pathway. It is known that APPL1 is a key scaffold protein in adiponectin signaling and its function is the recruitment of several proteins, such as MEK, downstream of the adiponectin receptor that have the capacity to transduce adiponectin signals [18]. Several studies have shown that lower APPL1 expression is related to decreases in GLUT4 translocation to the cell membrane and glucose uptake and a high rate of oxidation of fatty acids [19]. This is part of the insulin-sensitizing mechanism of adiponectin in cells. On the other hand, it is known that APPL2 is a regulatory protein that has opposite effects to APPL1, decreasing adiponectin signaling and reducing insulin sensitivity in target cells [20]. The data of the present work indicate that oral MTF treatment in the group of women with IR increased APPL1 and reduced APPL2 levels in the endometrial tissue of these women, independent of their PCOS condition, indicating an insulin-sensitizing action of MTF through adiponectin signaling, probably favoring the insulin-sensitizing endogen action of adiponectin in the cells. Interestingly, a study has shown that MTF (or insulin) inhibits the development of hypoadiponectinemia and prevents the downregulation of APPL1 in mesenteric resistance arteries [36]. Another study has shown that MTF (and adiponectin) can regulate the subcellular location of APPL1 and APPL2 and prevent the interaction between both molecules, favoring adiponectin signaling activation in C2C12 myotubes [20,37].

As already mentioned, the PCOS condition, besides obesity, promotes a pro-inflammatory environment in these women. Several reports have demonstrated that TNFα is one of the main antagonists of adiponectin action [10,38,39], and in the case of the endometria obtained from women with obesity, IR and PCOS, a high level of this pro-inflammatory cytokine has been detected [7]. In addition, endometrial cell cultures under hyperandrogenic and hyperinsulinemic conditions and stimulated with TNFα have a reduced capacity for glucose uptake [7]. This effect of TNFα is related to decreased expression of GLUT4 and of molecules involved in adiponectin (including adiponectin molecules) and insulin signaling in endometrial cells under these conditions [10,40].

Several molecules have been found to be involved in TNFα action to reduce insulin signaling in target cells, such as NF-κB and p38-MAPK proteins [7,10,32,34]. NF-κB activation has been associated with a reduction in adiponectin levels by TNFα action in human endometrial cells [10], which could explain, in part, the failures in the normal functioning of the endometrium, a tissue highly dependent on glucose for energy. On the other hand, the activation of p38-MAPK by TNFα has been mainly related to the mitogenic effects of TNFα, which could be associated with the higher risk of women with PCOS for developing an endometrial cancer [41,42,43]. It is relevant that in the present investigation it was found that treatment with MTF decreases both proteins, NF-κB and phosphorylated p38-MAPK in endometrial tissues. This indicates that MTF could improve insulin signaling by different pathways: directly, by increasing levels of molecules involved in the insulin signaling pathway, and indirectly, by regulating the adiponectin and TNFα signaling pathways. In this case, MTF could enhance the insulin-sensitizing effect of adiponectin and/or decrease the levels of molecules related to the TNFα signaling pathway and therefore diminish the insulin-repressor effect of this pro-inflammatory cytokine associated with obesity. It is possible that the positive effects of MTF on endometrial cells, independent of a PCOS condition, are due primarily to the effect of MTF on TNFα signaling in these cells. The latter agrees with another in vitro study from our laboratory [10], which shows that TNFα alone or TNFα plus insulin and testosterone (to mimic a PCOS environment) can increase protein levels of molecules involved in the TNFα signaling pathway and can decrease levels of molecules associated with adiponectin signaling, besides levels of GLUT4 in the endometrial cells.

The present study shows, for the first time, that traditional treatment with MTF for an IR condition is able to re-establish the action of endogen insulin-sensitizing molecules, such as adiponectin, in endometrial tissue under pathological conditions, this being the effect of MTF independent of the PCOS condition. In previous investigations carried out by our group, it was observed that an obesity state could exacerbate the failures associated with insulin action in endometrial tissue from women with PCOS [7,10,40].

Therefore, in the present work, the obesity condition was also evaluated in an in vitro model using human endometrial cells by adding TNFα to the cell cultures. Several studies have shown that TNFα alters the expression and activation of molecules involved in the MTF signaling pathway, such as AMPK, to induce GLUT4 expression [10,40]. The activation of AMPK is involved not only in MTF signaling but is also related to the action of adiponectin in improving insulin action [11,12,13,14]. These observations agree with the negative effects of TNFα on other molecules associated with the adiponectin pathway (such as adiponectin molecules and APPL1) and the reduction of GLUT4 levels and glucose uptake in human endometrial cells [7,10]. Cabrera-Cruz et al. 2020 [40] showed that AMPK activation, GLUT4 levels and glucose uptake reduced by TNFα in human endometrial cells may be prevented by treatment with an endogenous insulin sensitizer called myo-inositol [40]. Interestingly, the positive effects of myo-inositol occur in the presence or absence of hyperandrogenic and hyperinsulinemic in vitro conditions, suggesting that the action of insulin-sensitizing molecules is effective in an obesity environment independent of PCOS. In this investigation, it has been shown that MTF could improve insulin signaling by increasing the levels of molecules involved in insulin (p-AS160) signaling and regulating molecules involved in adiponectin signaling (APPL1, APPL2 and p-MEK) in endometrial cells under TNFα conditions intended to emulate the obesity condition. Therefore, the present study shows that MTF is capable of not only inducing the expression of molecules involved in insulin and adiponectin signaling pathways but also counteracting the negative effects of TNFα on endometrial tissue, thus improving the action of adiponectin and insulin in these cells under pathological conditions. Moreover, in the in vitro model it was evaluated whether other insulin-sensitizing molecules, such as the adiponectin agonist, can re-establish the TNFα-induced diminution of APPL1 and increase APPL2 levels in endometrial cells. In fact, the results show that the adiponectin agonist increased APPL1 and reduced APPL2 levels in cells under TNFα stimulus. Furthermore, the adiponectin agonist is able to re-establish the levels of molecules related to insulin signaling, such as p-AS160 and IRS1, in endometrial cells treated with TNFα. In addition, this study shows that the adiponectin agonist can increase the active form of IRS1 (p-IRS1 Y612), with no change in the inactive form of IRS1 (p-IRS1 S270). This is in accordance with a previous study where both forms of IRS1 were altered in the endometria from women with both obesity and PCOS compared with women with normal weight or with obesity [32]. Furthermore, it is known that TNFα can affect the insulin pathway by promoting either the activation or the inactivation of IRS1, in this case by being phosphorylated on serine residues due to inactivating kinases induced by TNFα. Importantly, in endometrial cells under hyperandrogenic and hyperinsulinemic conditions, TNFα increases the inactive form of IRS1, suggesting that a PCOS environment induces different alterations of p-IRS1 compared with the obesity condition in this in vitro model [32]. Here, it was observed that TNFα alone (without PCOS conditions) induces a decrease the active p-IRS1 levels, whereas the adiponectin agonist can re-establish basal levels. Thus, in the presence of a pro-inflammatory molecule that resembles an obesity condition, the action of insulin sensitizers can re-establish the expression of molecules to improve insulin signaling in endometrial cells.

In conclusion, this study shows that one of the mechanisms by which MTF could improve insulin sensitivity is by favoring the signaling pathway of an endogenous insulin-sensitizing molecule, such as adiponectin, increasing the expression of APPL1 and decreasing that of APPL2. On the other hand, all the evidence indicates that the action of insulin sensitizing molecules, either exogenous molecules, such as MTF (or myo-inositol as in other studies), or endogenous molecules, such as adiponectin, can improve insulin sensitivity in endometrial cells in pathological environments related to obesity-associated IR in the presence or absence of PCOS. Knowing the molecules that vary their expression due to the effect of these insulin-sensitizing molecules could help to better understand the failures observed in the endometria of women with obesity and PCOS. These data could also help us to develop new treatments focused on target molecules or improve known treatments that can normalize endometrial function in these women.

Finally, these results indicate that standard treatment of obesity-associated systemic IR may also improve a local condition of IR observed in the endometria of women with PCOS, improving endometrial function and the reproductive failures present in these women.

## Figures and Tables

**Figure 1 ijms-23-03922-f001:**
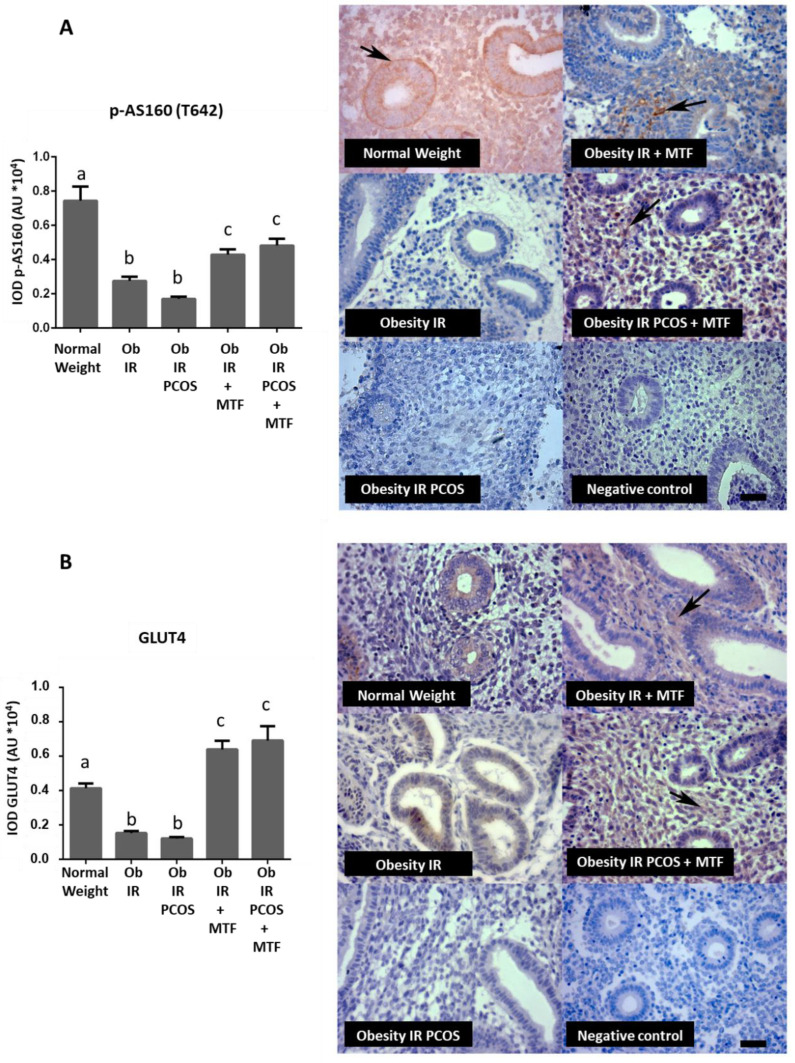
Effect of metformin on expression levels of molecules associated with insulin signaling in endometria from women with insulin resistance, obesity and PCOS. p-AS160 (T642) (**A**) and GLUT4 (**B**) levels in endometria from the study groups were determined by IHC (*n* = 7 samples in each group): Normal Weight group (control group), Obesity + IR group, Obesity + IR + PCOS group, Obesity + IR group treated with metformin (MTF) and Obesity + IR + PCOS group treated with MTF. MTF treatment: 850 mg twice a day for at least 12 weeks. Photomicrographs show the tissue location of p-AS160 (T642) (in (**A**)) and GLUT4 (in (**B**)) by immune-positive brown color staining in each sample (arrows). The graphs show the semi-quantification of each protein level in endometrial tissue obtained by the IOD tool. Negative control: tissue samples in the absence of primary antibody. Bar = 50 μm; images at 400×. In the graph, a ≠ b ≠ c indicates a statistically significant difference with a *p*-value < 0.05; non-parametric test and Dunn’s post hoc test. Data are shown as mean AU (arbitrary units *10^4^) ± SEM.

**Figure 2 ijms-23-03922-f002:**
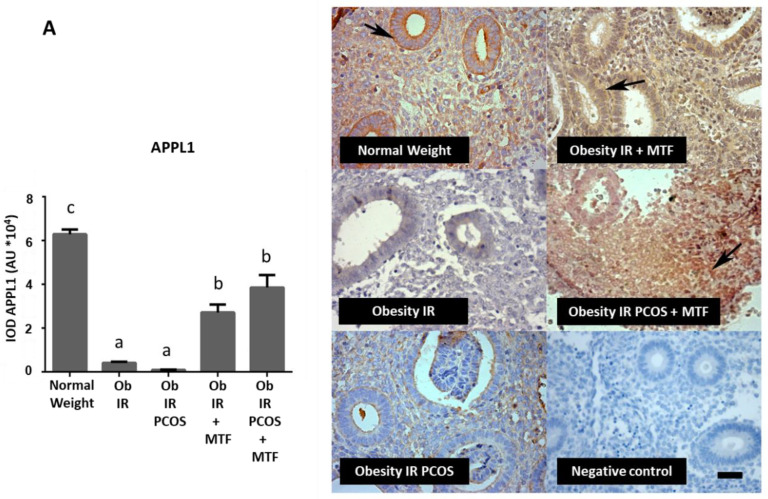
Effect of metformin on expression levels of molecules associated with adiponectin signaling in endometria from women with insulin resistance, obesity and PCOS. APPL1 protein (**A**) and APPL2 protein (**B**) levels in endometria from the study groups were determined by IHC (*n* = 7 samples in each group): Normal Weight group (control group), Obesity + IR group, Obesity + IR + PCOS group, Obesity + IR group treated with metformin (MTF) and Obesity + IR + PCOS group treated with MTF. MTF treatment: 850 mg twice a day for at least 12 weeks. Photomicrographs show the tissue location of APPL1 (in (**A**)) and APPL2 (in (**B**)) by immune-positive brown color staining in each sample (arrows). The graphs show the semi-quantification of each protein level in endometrial tissue obtained by the IOD tool. Negative control: tissue samples in the absence of primary antibody. Bar = 50 μm; images at 400×. In the graph, a ≠ b ≠ c indicates a statistically significant difference with a *p*-value < 0.05; non-parametric test and Dunn’s post hoc test. Data are shown as mean AU (arbitrary units *10^4^) ± SEM.

**Figure 3 ijms-23-03922-f003:**
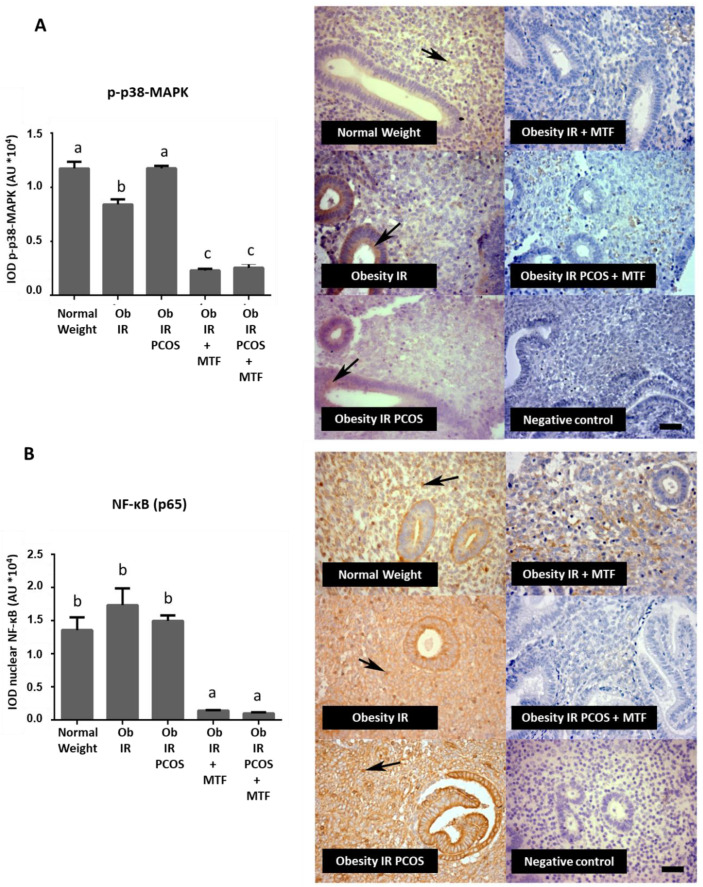
Effect of metformin on expression levels of p38-MAPK in its active form and NF-κB in the endometria of women with insulin resistance, obesity and PCOS. p-p38-MAPK (T180/Y182) protein (active form) (**A**) and NF-κB (p65 subunit) protein (**B**) levels in endometria from the study groups were determined by IHC (*n* = 7 samples in each group): Normal Weight group (control group), Obesity + IR group, Obesity + IR + PCOS group, Obesity + IR group treated with metformin (MTF) and Obesity + IR + PCOS group treated with MTF. MTF treatment: 850 mg twice a day for at least 12 weeks. Photomicrographs show the tissue location of APPL1 (in (**A**)) and APPL2 (in (**B**)) by immune-positive brown color staining in each sample (arrows). The graphs show the semi-quantification of each protein level in endometrial tissue obtained by the IOD tool. Negative control: tissue samples in the absence of primary antibody. Bar = 50 μm; images at 400× (in (A): image of 200× in inner box). In the graph, a ≠ b ≠ c indicates a statistically significant difference with a *p*-value < 0.05; non-parametric test and Dunn’s post hoc test. Data are shown as mean AU (arbitrary units *10^4^) ± SEM.

**Figure 4 ijms-23-03922-f004:**
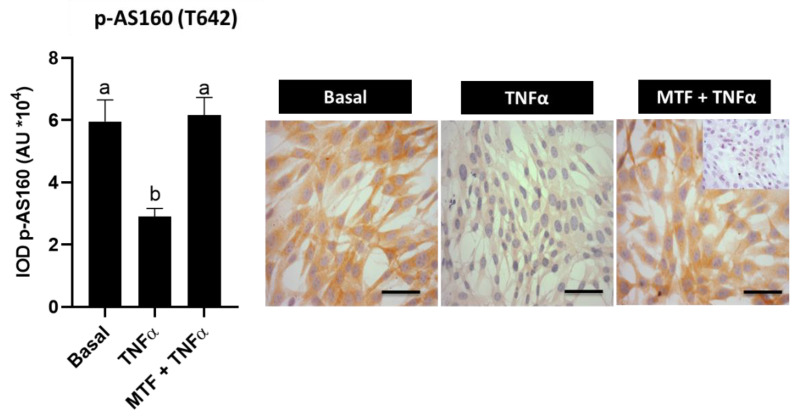
Effect of metformin on the expression of phosphorylated AS160 molecule in human endometrial cells exposed to TNFα treatment. p-AS160 (T642) protein levels in human endometrial cells treated for 24 h with TNFα (100 ng/mL) with or without metformin (MTF, 20 nM) by immunocytochemistry (ICC). Photomicrographs show the tissue location of p-AS160 by immune-positive brown color staining in each sample. The graphs show the semi-quantification obtained by the IOD tool. Inner box: negative control; bar = 50 μm; images at 400× (in (A): image of 200× in inner box). Three independent experiments were performed for each treatment (*n* = 3; in duplicate). In the graph, a ≠ b indicates a statistically significant difference with a *p*-value < 0.05; non-parametric test and Dunn’s post hoc test. Data are shown as mean AU (arbitrary units *10^4^) ± SEM.

**Figure 5 ijms-23-03922-f005:**
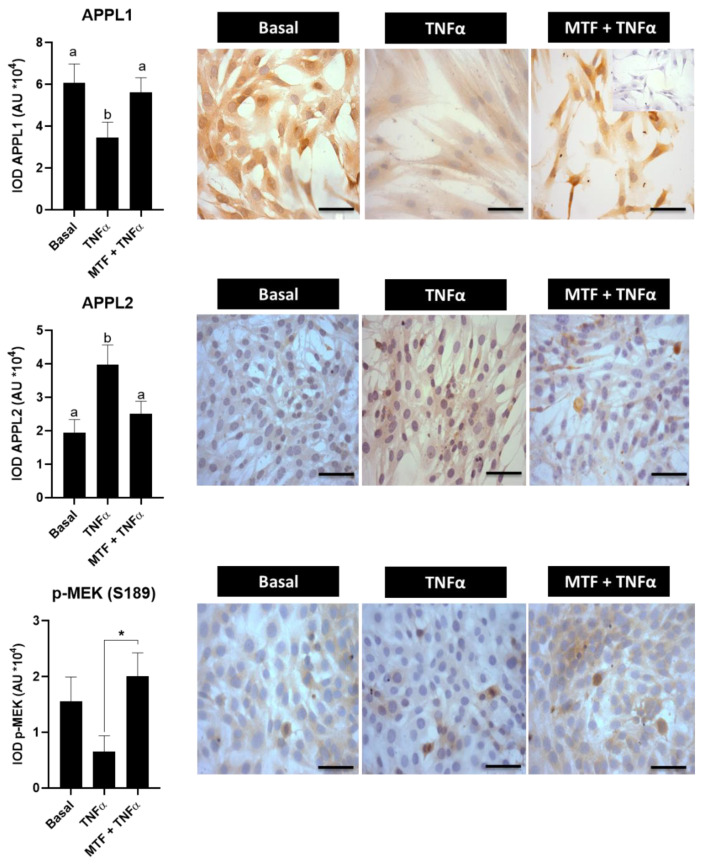
Effect of metformin on the expression of adiponectin signaling-associated molecules in human endometrial cells exposed to TNFα treatment. APPL1, APPL2 and phosphorylated MEK (S189) protein levels in human endometrial cells treated for 24 h with TNFα (100 ng/mL) with or without metformin (MTF, 20 nM) as determined by immunocytochemistry (ICC). Photomicrographs show the tissue locations of APPL1, APPL2 and p-MEK (S189) by immune-positive brown color staining in each sample. The graphs show the semi-quantification obtained by the IOD tool. Inner box: negative control; bar = 50 μm; images at 400× (in (A): image of 200× in inner box). Three independent experiments were performed for each treatment (*n* = 3; in duplicate). In the graphs, a ≠ b indicates a statistically significant difference with a * *p*-value < 0.05. Regarding the p-MEK graph: *p*-value < 0.05 TNFα vs. TNFα+MTF; non-parametric test and Dunn’s post hoc est. Data are shown as mean AU (arbitrary units *10^4^) ± SEM.

**Figure 6 ijms-23-03922-f006:**
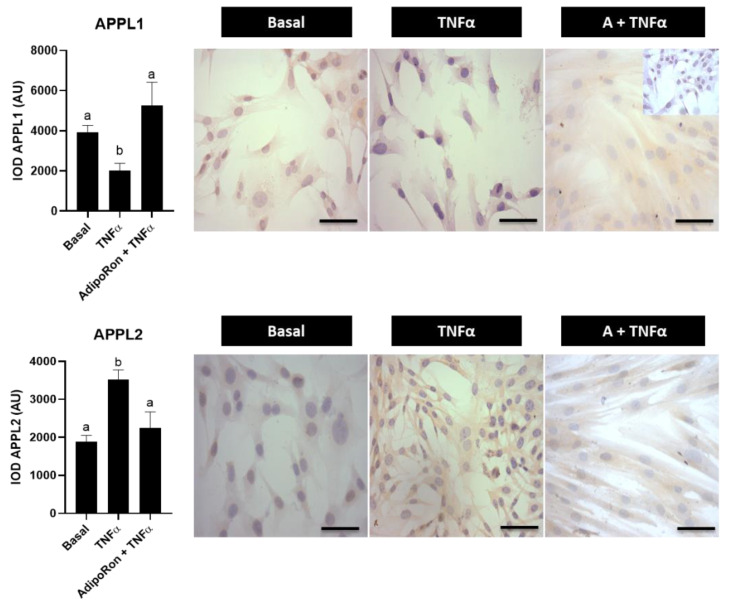
Effect of AdipoRon on the expression of adiponectin signaling-associated molecules in human endometrial cells exposed to TNFα treatment. APPL1 and APPL2 protein levels in human endometrial cells treated for 24 h with TNFα (100 ng/mL) with or without AdipoRon (A, adiponectin agonist) (30 µM) by immunocytochemistry (ICC). Photomicrographs show the tissue locations of APPL1 and APPL2 by immune-positive brown color staining in each sample. The graphs show the semi-quantification obtained by the IOD tool. Inner box: negative control; bar = 50 μm; images at 400× (in (A): image of 200× in inner box). Three independent experiments were performed for each treatment (*n* = 3; in duplicate). In the graphs, a ≠ b indicates a statistically significant difference with a *p*-value < 0.05. Non-parametric test and Dunn’s post hoc test. Data are shown as mean AU (arbitrary units) ± SEM.

**Figure 7 ijms-23-03922-f007:**
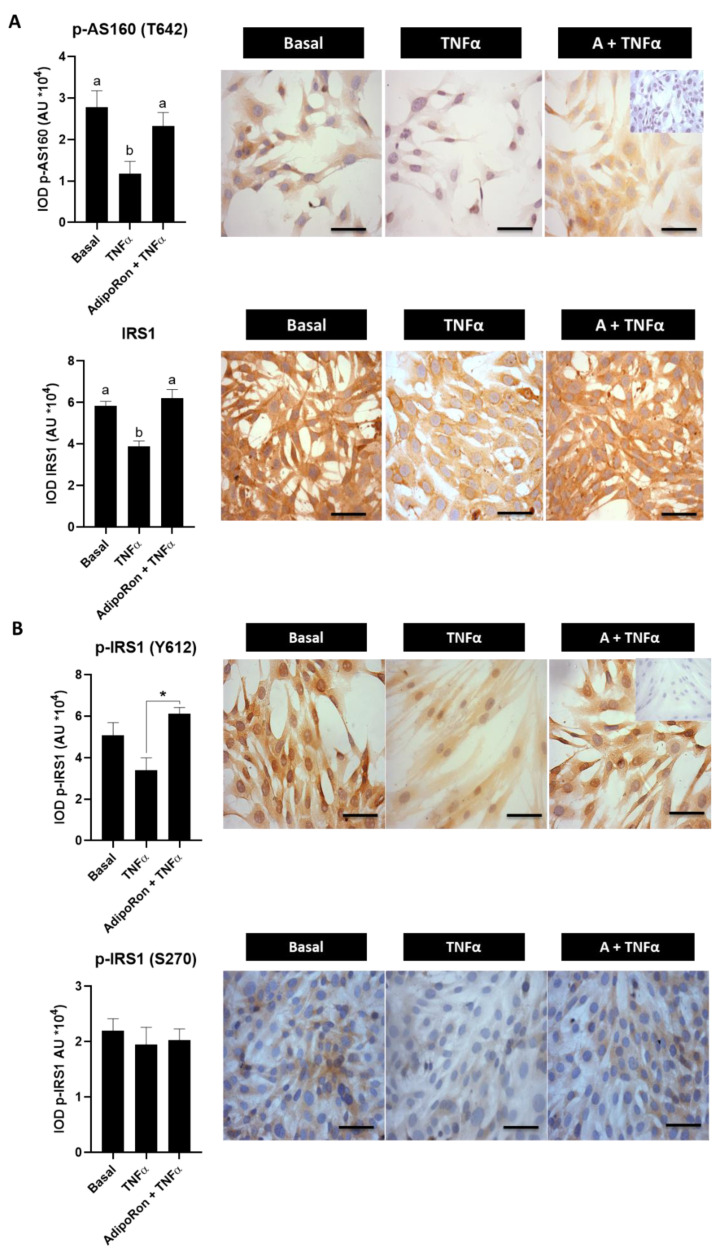
Effect of AdipoRon on the expression of insulin signaling-associated molecules in human endometrial cells exposed to TNFα treatment. p-AS160 (T642), IRS1, p-IRS1 (Y612) and p-IRS1 (S270) protein levels in human endometrial cells treated for 24 h with TNFα (100 ng/mL) with or without AdipoRon (A, adiponectin agonist) (30 µM) by immunocytochemistry (ICC). (**A**) Photomicrographs show the tissue location of p-AS160 (T642) and IRS1 protein by immune-positive brown color staining in each sample. (**B**) Photomicrographs show the tissue location of p-IRS1 (Y612) and p-IRS1 (S270) proteins by immune-positive brown color staining in each sample. The graphs show the semi-quantification obtained by the IOD tool. Inner box: negative control; bar = 50 μm; images at 400× (in (A): image of 200× in inner box). Three independent experiments were performed for each treatment (*n* = 3; in duplicate). In the graphs, a ≠ b indicates a statistically significant difference with a *p*-value < 0.05 (in (**A**)). Regarding p-IRS1 (Y612) (in (**B**)): * *p*-value < 0.05 TNFα vs. TNFα + A; non-parametric test and Dunn’s post hoc test. Data are shown as mean AU (arbitrary units *10^4^) ± SEM.

**Table 1 ijms-23-03922-t001:** Clinical characteristics of study groups.

Parameters	Normal Weight	Obesity + IR	Obesity + IR + PCOS
Age (years)	26.6 ± 5.6	26.9 ± 3.2	25.9 ± 2.4
BMI (kg/m^2^)	22.4 ± 2.1	34.2 ± 2.2 ^&^	32.6 ± 3.4 ^&^
Estradiol (pmol/L)	168.7 ± 77	159.5 ± 58	143.6 ± 40
Testosterone (ng/dL)	36.6 ± 10.7	31.8 ± 9.6	49.4 ± 9.2 *
Androstenedione (ng/mL)	4.9 ± 1.3	4.5 ± 1.7	5.3 ± 1.4 *
SHBG (nmol/L)	68.9 ± 0.8	52.0 ± 1.8	25.7 ± 11 *
FAI	1.9 ± 0.6	2.1 ± 0.9	6.7 ± 1.9 *
HOMA-IR	1.49 ± 1.1	4.1 ± 0.7 ^&^	4.5 ± 0.9 ^&^
ISI Composite	8.1 ± 4.7	2.4 ± 1.6 ^&^	2.09 ± 1.1 ^&^

The values are expressed as means ± SE. PCOS, polycystic ovary syndrome; BMI, body mass index (Obesity: BMI ≥ 30); SHBG, sex hormone binding globulin; FAI, free androgen index (Normal FAI < 4.5); HOMA-IR, homeostatic model assessment for insulin resistance (Normal HOMA-IR < 2.6); ISI Composite, insulin sensitivity index of Matsuda (Normal ISI-Composite > 3). * *p*-value < 0.05 Obesity + IR + PCOS vs. other groups; ^&^
*p*-value < 0.05 Obesity + IR and Obesity + IR + PCOS vs. Normal Weight.

## Data Availability

Not applicable.

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
