# Peer review of "Metformin Treatment Regulates the Expression of Molecules Involved in Adiponectin and Insulin Signaling Pathways in Endometria from Women with Obesity-Associated Insulin Resistance and PCOS"

_ijms, 2022, doi:10.3390/ijms23073922_

Round 1

Reviewer 1 Report

I consider this manuscript to be well structured and developed. However, authors should be important changes to made for publishing the present study:

Figure 1A: Only half “Obesity IR PCOS” and “Negative Control” images are shown, and they don`t have their name over the image.

Figure 1B: Review “Obesity IR PCOS” and “Negative Control” images. They look like several superimposed images.

Figure 1: In the caption " a ≠ b ≠ c " appears, but "c" does not appear in the graphs.

Figure 3: Images and caption are the same as “figure 2”.

Results. Metformin re-established the levels of insulin and adiponectin signaling molecules in cultured human endometrial cells: I consider that the last paragraph (“These results suggest that MTF treatment can improve the obesity-induced molecular alterations related to insulin and adiponectin (an insulin-sensitizer) in human endometrial cells.”) belongs to the section "Discussion".

Figure 4: Images and graphs are not shown.

Figure 5: In the caption appear “Regarding p-MEK graph: *P value <0.05 MTF versus TNFα + MTF.” It should be “TNFα versus TNFα + MTF.”

Figure 7A: I recommend changing the images of this figure because the staining is poorly observed. Also, the “TNFα” image requires a blank before taking the picture so it doesn't look so blue.

Author Response

Point 1: Figure 1A: Only half “Obesity IR PCOS” and “Negative Control” images are shown, and they don`t have their name over the image.

Response 1: Thank you for the observation. In the original version of the manuscript the images had shifted position, but in the revised version this was corrected.

Point 2: Figure 1B: Review “Obesity IR PCOS” and “Negative Control” images. They look like several superimposed images.

Response 2: The images were technically reviewed and only one microphotograph compose it.

Point 3: Figure 1: In the caption " a ≠ b ≠ c " appears, but "c" does not appear in the graphs.

Response 3: Thank you for the observation. The missing letter was added in the image as it corresponds to the statistical analysis.

Point 4: Figure 3: Images and caption are the same as “figure 2”.

Response 4: Thank you, Figure 3 has already been corrected in the revised vesion.

Point 5: Results. Metformin re-established the levels of insulin and adiponectin signaling molecules in cultured human endometrial cells: I consider that the last paragraph (“These results suggest that MTF treatment can improve the obesity-induced molecular alterations related to insulin and adiponectin (an insulin-sensitizer) in human endometrial cells.”) belongs to the section "Discussion".

Response 5: We agree with the Reviewer’ s comment. The paragraph will be removed from the Results section, and it has been included in the Discussion section of the revised version.

Point 6: Figure 4: Images and graphs are not shown.

Response 6: In the original version of the manuscript, the images lost their correlative order. In the new and improved version it is possible to observe the images accordingly.

Point 7: Figure 5: In the caption appear “Regarding p-MEK graph: *P value <0.05 MTF versus TNFα + MTF.” It should be “TNFα versus TNFα + MTF.”

Response 7: Thank you for the observation. This point has already been corrected in the new version.

Point 8: Figure 7A: I recommend changing the images of this figure because the staining is poorly observed. Also, the “TNFα” image requires a blank before taking the picture so it doesn't look so blue.

Response 8: The images have already been changed in the revised version. Thank you very much for your recommendation.

Reviewer 2 Report

Presented manuscript is very interesting and also important, however, reviewer has few suggestions for Authors:

1. Page 2, line 8. It is "...of GLUT4 vesicles to the cellular plasma membrane". I think that it should be "... of GLUT4 vesicles from the intracellular compartment (space) to the cellular plasma membrane"

2. Page 2, paragraph 3, line 15. It is "... cell peripher.y..." > "... cell membrane..."

3. Page 2, in last paragraph is used "non-insulin-dependent diabetic patients" Better version is "type 2 diabetic patients, or patients with type 2 diabetes" 

4. Page 9. It ijs "... reduces GLUT4". It is uncelar. What is reduced? Expression of GLUT4 in cell, or its level in plasma membrane, or maybe its transcription and translation. 

Page 15. "myoinositol" should be write "myo-inositol" (few times).

I think that abbreviations used in manuscript, such as AMPK, APPL, TNF and so on need explanation.

Author Response

Point 1: Page 2, line 8. It is "...of GLUT4 vesicles to the cellular plasma membrane". I think that it should be "... of GLUT4 vesicles from the intracellular compartment (space) to the cellular plasma membrane"

Response 1: Thank you for the suggestion. The line was changed in the new version of the manuscript (page 2 , line 8).

Point 2: Page 2, paragraph 3, line 15. It is "... cell peripher.y..." > "... cell membrane..."

Response 2: It has been changed according to the Revewer’s suggestion (page 2).

Point 3: Page 2, in last paragraph is used "non-insulin-dependent diabetic patients" Better version is "type 2 diabetic patients, or patients with type 2 diabetes" 

Response 3: Thank you, this suggestion has been included in the revised version of the manuscript (page 2, last paragraph).

Point 4: Page 9. It ijs "... reduces GLUT4". It is uncelar. What is reduced? Expression of GLUT4 in cell, or its level in plasma membrane, or maybe its transcription and translation. 

Response 4: Thank you for the observation. In the condition of obesity, protein levels of GLUT4, among other proteins, are reduced. This point has already been incluided in the new version of the manuscript (page 10).

Point 5: Page 15. "myoinositol" should be write "myo-inositol" (few times).

Response 5: you Thank for the observation. The suggestion was considered (myo-inositol was changed to myo-inositol throughout the text).

Point 6: I think that abbreviations used in manuscript, such as AMPK, APPL, TNF and so on need explanation.

Response 6: Thank you for your comment, it was considered in the new version of the manuscript.

Reviewer 3 Report

ijms-1635194

Metformin treatment regulates the expression of molecules involved in adiponectin and insulin signaling pathways in endometrium from women with obesity-associated insulin resistance and PCOS.

The group Orostica and Vega examined the potential molecular implications of metformin treatment on the endometrium of obese women with PCOS and found evidence that metformin can improve markers of insulin resistance and adiponectin signaling. The data presented is a founded on their previous works and offers additional insights on the metabolic underpinnings of PCOS. Although the data is compelling, there are some aspects of the study that needs improvements to be deemed publishable.

  1. Immunohistochemistry. Details are needed to how many fields were assessed per sample to avoid bias. As a rule of thumb, 5-10 images within the tissue section are sufficient to address this issue. This also applies to the in-vitro cell culture experiments.

  1. Cell cultures and treatments: A critical issue here is that the cells were only treated with Tnf-a - thus the condition only mimic obesity and not PCOS. Cells should have been incubated with testosterone in addition to Tnf-a.

  1. Statistical evaluation. Non-parametric analysis was used. What is the reason why non-parametric tests were used? For the in-vitro studies, there are more than three groups and thus Mann-Whitney is not applicable.

  1. Results. Figures (bar graphs) are confusing. Some bars have letters on top of them and some did not have any. I suggest that the authors put letters on top of all the bars to indicate differences/no difference between groups.

Arrows point on different parts of the tissue. Some are pointing on endometrial glands and some on endometrial cells. Is there a difference in the localization of these markers in relation to the observed improvements ex-vivo?

Figure 2 and Figure 3 are the same.

To show whether the action of metformin is independent of obesity, it would be important to incorporate (in future studies) samples from lean women with PCOS.

Metformin is shown to increased circulating adiponectin and decreased C-reactive protein and TNF-a in women with PCOS. Are the improved tissue markers ex-vivo due to this effect of metformin in-vivo?

In the in-vitro studies, how were the signals normalized? There are some slides with different number of cells which can confound the intensity of the signals.

  1. Discussion: Authors claim that their results show that there was an improved insulin action. This is a speculation as there is no experiment that treated the cells with exogenous insulin.

Minor comments:

The authors have to consult an English language editor as there are numerous statements that are grammatically incorrect.

Author Response

Point 1: Immunohistochemistry. Details are needed to how many fields were assessed per sample to avoid bias. As a rule of thumb, 5-10 images within the tissue section are sufficient to address this issue. This also applies to the in-vitro cell culture experiments.

Response 1: Thank you for the observation. This important point was included in the Materials and Methods section (pages 4 and 5, for both techniques).

Point 2: Cell cultures and treatments: A critical issue here is that the cells were only treated with Tnf-a - thus the condition only mimic obesity and not PCOS. Cells should have been incubated with testosterone in addition to Tnf-a.

Response 2: Thank you for your comment. In the results section it is indicated that the main objective was to partially resemble the inflammatory condition associated with obesity. This, supported by the results of other studies associated with obesity from our laboratory [ref. 7 from the manuscript]. Interestingly, another in-vitro study from our laboratory shows that TNFa alone or TNFa plus insulin and testosterone (to mimic PCOS environment) can increase protein levels of molecules involved in its signaling pathway and can decrease of adiponectin signaling molecules expression and GLUT4 levels in the endometrial cells [ref. 10 from the manuscript]. This supports what was seen in the present study: the effect of MTF observed in the tissue samples was independent of PCOS. This point was incluided in Discussion section (first paragraph, page 16).

Point 3: Statistical evaluation. Non-parametric analysis was used. What is the reason why non-parametric tests were used? For the in-vitro studies, there are more than three groups and thus Mann-Whitney is not applicable.

Response 3: Thank you for your comment. The statistics were performed with the non-parametric Kruskal Wallis test to evaluate significant differences in the total of our data, since the distribution of the data of our numerical variables is not normally distributed, according to the Shapiro-Wilk test. Then, a test was made to compare between the different groups. The Mann-Whitney test was not used. This section has already been corrected in the new version of the manuscript (page 5).

Point 4: Results. Figures (bar graphs) are confusing. Some bars have letters on top of them and some did not have any. I suggest that the authors put letters on top of all the bars to indicate differences/no difference between groups.

Response 4: We agree with this observation. The graphics were corrected and the letters were added according to the statistics in the new version.

Point 5: Arrows point on different parts of the tissue. Some are pointing on endometrial glands and some on endometrial cells. Is there a difference in the localization of these markers in relation to the observed improvements ex-vivo?

Response 5: Thank you for the observation. An analysis differentiating the glands from the endometrial stroma was not performed. In this case, the arrows indicate (as an example) the brown positive staining that represents the presence of the protein. Therefore, in the Figure legends of the revised version this point has been clarified.

Point 6: Figure 2 and Figure 3 are the same.

Response 6: Sorry for that error. In the original version of the manuscript the images have shifted position, but in the revised version this was corrected. Figure 3 has already been corrected.

Point 7: To show whether the action of metformin is independent of obesity, it would be important to incorporate (in future studies) samples from lean women with PCOS.

Response 7: Thank you very much for your suggestion. In fact, previous studies from our laboratory included samples from normal weight women with PCOS. However, being a rare condition in our country (Chile), these few samples were not included in the present study. As stated, in our previous investigation it was observed that some metabolic parameters were altered in these women (normal-weight PCOS women), but the effect was always greater when the women presented both conditions, PCOS and Obesity [ref. 7 from the manuscript].

Point 8: Metformin is shown to increased circulating adiponectin and decreased C-reactive protein and TNF-a in women with PCOS. Are the improved tissue markers ex-vivo due to this effect of metformin in-vivo?

Response 8: Thank you for your comment. It is likely that the effect of MTF is multifactorial and that it is related to the decrease in inflammatory parameters, such as TNF and C-reactive protein. This will be included in the Discussion section (first paragraph, page 15).

Point 9: In the in-vitro studies, how were the signals normalized? There are some slides with different number of cells which can confound the intensity of the signals.

Response 9: We agree and thank your observation. All analyzes are performed using photomicrographs taken at the same optical magnification (400x). Although we did not always obtain the same confluence, the protein levels were compared with the same number of cells analyzed per experiment (per graph). We were careful to do the analyzes for a particular protein in wells that had the same cell confluence.

Point 10: Discussion: Authors claim that their results show that there was an improved insulin action. This is a speculation as there is no experiment that treated the cells with exogenous insulin.

Response 10: Thank you for your comment. We performed glucose uptake assays with these MTF-treated cells with or without TNF-alpha. However, the results provided too much dispersion and some data even turned out to be inconsistent. We assumed that the problem was technical and we were unable to add them to this study. Due to economic issues and the pandemic, it was impossible to repeat these experiments. In the discussion section, the expression "insulin action" was modified and what was improved by the effect of MTF was better explained.

Point 11: Minor comments: The authors have to consult an English language editor as there are numerous statements that are grammatically incorrect.

Response 11: Thanks for the necessary recommendation. The revised manuscript have been reviewed by a native english speaker with expertise in this scientific area.

Round 2

Reviewer 3 Report

Minor Comments

  1. Despite the manuscript being read and edited by a native English speaker, I find the readability of the manuscript still needs improvement.
  2. What is the difference between the letters (Fig 7A) and astesisk in Fig 7B? 

Author Response

Thank you for your comments. Different letters in Figure 7A indicate statistically significant differences between the bars. In figure 7B the asterisk indicates a statistically significant difference only between TNFa and AdipoRon + TNFa. The graph below (7B) did not present a significant difference between the different treatments. The legend of Figure 7 explains better the meaning of the letters and the asterisk in Figure A and B, respectively.